# College students' sense of belonging and alcohol use amidst COVID-19: Evidence from a 21-day daily diary study

**Maithreyi Gopalan**[1]*, Jilli Jung[1], Chiang Shou-Chun[2], Ashley Linden-Carmichael[3], Stephanie Lanza[3]

**1** Department of Education Policy Studies, The Pennsylvania State University, University Park, PA, United States of America, **2** Department of Human Development and Family Sciences, Texas Tech University, Lubbock, Texas, United States of America, **3** Edna Bennett Pierce Prevention Research Center, The Pennsylvania State University, University Park, PA, United States of America

* smg632@psu.edu

**Data Availability Statement:** Data cannot be shared publicly because of the research protocol and IRB rules. Data are available from the authors upon request, who will facilitate Institutional Data

## Abstract

### Objective

Alcohol use, combined with the heightened mental health crisis among college students highlighted during the pandemic, remains a significant public health concern. We examine (1) how college students' daily assessed sense of *belonging* with their institution, a key protective factor for better collegiate mental health, is associated with same-day alcohol-use behaviors (2) and how the associations are moderated by key sociodemographic characteristics relevant to this population (women, minoritized students, first-generation [FG], and students identifying as Lesbian, Gay, Bisexual, Transgender, or Queer [LGBTQ]) amidst COVID-19.

### Method

We used mixed models using data from a 21-day daily diary study of college students ($N = 2,012$) in Spring 2021.

### Results

Results revealed that on days when students felt more uncertainty about their sense of belonging to their college (i.e., low belonging), they were less likely to drink, and drink less overall. This effect was observed after students were back on campus after pandemic-related college closures ended. Heterogeneity by minoritized student subgroups were also observed.

### Conclusions

College students' sense of belonging continues to be an important psychosocial determinant of health and health behaviors among young adults; at times in unintended ways. This reiterates the importance of examining dynamic relationships between belonging and population health.

Access and Ethics Committee approvals from Penn State IRB for researchers who meet the criteria for access to confidential data. The non-author contact for the data access committee related to this project is Cortney Whetzel (cae130@psu.edu).

**Funding:** SL acknowledges funding for the survey administration from the Social Science Research Institute, Huck Institutes of the Life Sciences, and the College of Health and Human Development at The Pennsylvania State University. The funders did not play any role in the study design, data collection and analysis, decision to publish, or preparation of the manuscript. All remaining errors are our own.

**Competing interests:** The authors have declared that no competing interests exist.

**Public health significance statements:** These results provide important insight into the linkages between a key psychosocial factor—students' sense of belonging in college—and their alcohol use patterns amidst COVID-19. Institutional programming and prevention efforts to curb alcohol misuse should be implemented with consideration of how those linkages may differ dynamically considering both between- and within-person variance in belonging.

## Introduction

Alcohol use among young adults aged 18–25, who exhibit higher rates of drinking than any other age group, is a significant and widespread public health concern [1]. For example, 19/ 20-year-olds who currently attend postsecondary universities and do not live with their parents exhibit the highest rates of drinking among adults in the US [12.4%; 2]. Several studies have focused on the harmful effects of drinking among college students; however, given the rapid shutdown of college campuses at the onset of the COVID-19 pandemic, how did alcohol use patterns evolve amidst prolonged social isolation and during their subsequent return to campus when campus closures ended?

Research on alcohol use patterns during the initial onset of the pandemic (i.e., early to mid-2020) revealed mixed patterns and trends. Specifically, while some studies found that college students reported heavier use during the pandemic (relative to prior to the pandemic) [3, 4], others found that young adults might be drinking *less* overall, but instead drinking more frequently, and primarily with their family and/or alone [5, 6]. More consistently, the pandemic also fueled a brewing mental health crisis across the age spectrum, but especially so for college students given the unprecedented disruption to higher education [7]. Given that major differences exist in the impact of COVID-19 on health outcomes by gender, minoritized student status, first-generation status, and LGBTQ status [8–11], examining heterogeneity in alcohol use patterns across various college student subgroups amidst COVID-19 remains important. For example, increases in alcohol frequency, heavy drinking, and alcohol-related harms were observed only in women [12], and the link between psychological distress and alcohol use and use severity was also observed only in women [13, 14]. Lastly, during the pandemic, individuals who identify as LGBTQ+ have been at disproportionate risk for mental health concerns and substance use [15, 16].

On the prevention side, a key emerging protective factor against adverse mental health outcomes, such as anxiety and depression, which often co-occur with alcohol use [e.g., 17], was students' sense of belonging with their university [18]. For example, college students who reported feeling a strong sense of belonging with their institution prior to the onset of the pandemic (e.g., "I feel like I belong at [college]"; "How often, if ever, do you wonder: Maybe I do not belong here?", reverse coded) reported lower levels of depressive symptoms and anxiety during the pandemic, even after controlling for their baseline (pre-COVID-19 pandemic) depressive and anxiety symptoms [18]. However, to date, only a few studies have examined the direct linkages between belonging and alcohol use in college.

A sense of belonging, often considered a fundamental human motivation [19], has consistently been associated with greater persistence, achievement, and social and academic integration on college campuses [20–23]. More recently, studies have also begun to show the buffering effects of belonging with regard to mental health outcomes such as depression and anxiety [24, 25]. Most of these studies measure belonging and belonging uncertainty using items like: "I feel that I am a part of [college]"; I feel like I belong at [college]; "Sometimes I

worry that I do not belong in [college]"). We adapted a similar, extensively used, belonging uncertainty measure to a daily diary context in our current study. These studies underscore the importance of examining linkages between students' sense of belonging with their college and alcohol use particularly amidst COVID-19 and beyond.

There is also mixed evidence on the sign, magnitude, and consistency of associations between belonging and alcohol use in non-pandemic times. Among college samples, alcohol is commonly used to fulfill their desire to fit in and gain social approval [26–30]. There is, thus, some reason to expect a positive association between belonging and alcohol use, particularly when related to social drinking that is quite common on college campuses. For example, drinking frequency has been previously associated with greater belonging in college [31]. Also, students who socialize more drink more with their peers to gain approval [32], and are likely to feel more accepted and connected with peers in college. Indeed, Hamilton and DeHart [33] found that students who reported higher levels of needing to belong at baseline were *more* likely to drink on days when they experienced negative interpersonal interactions.

Importantly, however, students' sense of belonging with their institutions is not static; it is dynamic [20, 25, 34] and may thus represent a key, within-person, moderator that can be intervened upon [20, 22]. A recent longitudinal study of the association between college students' alcohol use and drinking norms specifically examined the moderating role of students' sense of belonging in college [29]. Findings indicated that students who perceived their peers as more approving of drinking consumed more alcohol and consumed alcohol more often– but only when their sense of belonging was lower than their own average levels of belonging. Yet, interventions targeting alcohol use among adolescents and young adults, even the most effective ones, rarely seem to consider this key social psychological factor in their designs [35].

While Neighbors and colleagues' work [29] documents the critical role of belonging in alcohol use among college students, there are several gaps in our knowledge that warrant additional research. First, there is consistent evidence of sociodemographic differences by race/ethnicity, gender, and first-generation college student status in students' sense of belonging. For example, women report higher sense of belonging with their campus than men, first-generation college students and minoritized students often report feeling lower belonging than their white and/or continuing-generation peers, especially in predominantly white institutions [20, 22, 24, 34]. The extent to which similar differences exist in the link between belonging and alcohol use remains unknown. Second, although longitudinal, Neighbors and colleagues' study of student perceptions of norms, self-reports of belonging, and alcohol use was conducted over two years in college with measures collected 8 times at 3-month intervals. However, college students' sense of belonging may vary day-to-day [20, 22, 34]. The current study aimed to examine whether day-to-day variations in students' belonging with their campus were associated with their alcohol use patterns.

## Current study

The overarching goal of the current study was to examine associations between intensively measured students' sense of belonging, assessed daily over 21 days, and reports of daily alcohol use and other key factors amidst COVID-19. Our daily-diary study ($N$ = 2,012) conducted in 2021 (amidst COVID-19 but after students had returned to campus after almost a year of remote/hybrid learning), will add nuance and clarity to the sign, strength, and consistency of associations between students' sense of belonging and alcohol use and inform outreach efforts for institutional programming. Additionally, by Spring 2021, when this study was conducted, this institution had transitioned back to pre-COVID-19 teaching modalities. The campus was open, and almost all classes had returned to in-person formats.

It is well-documented that college students' sense of belonging plays an important role in mental health and wellbeing outcomes [24, 25] and that sense of belonging is a dynamic process that may vary across long periods of time through college [29]. Given the limited work examining event-level predictors of alcohol use [36], including daily variations in belonging, the current study sought to address this gap using 21-day diary data from a large college student sample. Furthermore, given the paucity of research on the unique risk and resiliency factors related to alcohol use among college students from marginalized backgrounds [37] and recent work documenting underrepresented students from minoritized racial-ethnic backgrounds report a lower sense of belonging than their peers [18, 24, 38, 39], this study examines moderations in belonging and alcohol use by important sociodemographic factors as well. Specifically, this study sought to address the following research questions:

(RQ1) What is the within-person association between daily sense of belonging with their institution and daily alcohol use behaviors (likelihood of drinking and total number of drinks) across 21 days amidst the pandemic during 2021? And, (RQ2): How do associations between belonging and alcohol use behaviors vary by gender and health disparity sociodemographic characteristics—such as first-generation [FG] college students, minoritized students, and students identifying as LGBTQ?

## Method

As described above, we use a 21-day, daily diary survey and a quantitative, longitudinal analysis (using mixed models) to examine the association between day-to-day variations in students' belonging with their campus and their alcohol use patterns.

### Participants

Undergraduate students were recruited from a large, suburban, public, postsecondary university in the Northeast region of United States from Spring and Fall 2021 to participate in an online daily diary study—called the Student Engagement, Learning and Flourishing (SELF) study. Participant demographics are presented in Table 1. A random sample of participants were recruited via e-mail invitations. Students first completed a screener to determine eligibility. To be eligible, participants must have been a 1st through 4th-year, full-time, undergraduate student and aged 18–24 years. No other exclusion criteria were considered.

### Procedure

If students were eligible, the survey routed the participants to an "informed consent" document link that explicitly stated the potential risks and benefits of participating in the research study. If the students consented to participate, they completed an online baseline survey and were immediately enrolled in the daily diary portion. Each day participants received a survey link to their email and cell phone (at 9:00 am with a reminder one hour later to participants who had not yet completed the day's survey) that assessed health behaviors and its correlates. Participants received $15 for the baseline survey and up to $67 for the daily surveys, with extra incentives for higher completion. The university's Institutional Review Board approved all ethical considerations including the informed consent language/documentation, data collection, and analysis plan of the current study.

### Measures

**Daily belonging.** Each day students were asked: "Yesterday, when you think about [college], how often, if ever, did you wonder: Maybe I don't belong here?" (*0 = Never, 1 = Hardly*

**Table 1. Descriptive statistics of analytic sample.**

| Variable | Person-level N (%) | # of Person-Days for Each Group of Students |
|---|---|---|
| Gender (% women) | 1,309 (68.43) | 21,910 |
| LGBTQ (%) | 320 (16.73) | 5,261 |
| Race/Ethnicity | | |
| Non-Hispanic White | 1,349 (70.52) | 22,736 |
| Non-Hispanic Black | 48 (2.51) | 618 |
| Hispanic | 150 (7.84) | 2,352 |
| Non-Hispanic Asian | 276 (14.43) | 4,278 |
| All Other race/Multiracial | 90 (4.70) | 1,438 |
| Age M (SD) | | |
| 18 | 392 (20.49) | 6,306 |
| 19 | 463 (24.2) | 7710 |
| 20 | 425 (22.22) | 7,093 |
| 21 | 438 (22.9) | 7,187 |
| 22 | 164 (8.57) | 2,689 |
| 23 | 23 (1.2) | 334 |
| 24 | 8 (0.42) | 103 |
| Year in College | | |
| 1st year (1–2 semesters) | 471 (24.62) | 7,444 |
| 2nd year (3–4 semesters) | 483 (25.25) | 8,041 |
| 3rd year (5–6 semesters) | 432 (22.58) | 7,290 |
| 4th year (7–8 semesters) | 463 (24.2) | 7,576 |
| 5th year ~ (9–11 semesters) | 59 (3.08) | 1,019 |
| missing | 5 (0.26) | 52 |
| First generation | 324 (16.94) | 26,212 |
| Total in analytic sample | 1,913 | 31,422 |

*Note*. Analytic sample for the *any alcohol use* model was reported here. Analytic sample sizes by subgroups vary slightly across models due to missing cases in dependent variables.

*ever*, *2 = Sometimes*, *3 = Frequently*, *4 = Always*). We use this single-item measure, which is referred to as belonging uncertainty in the literature [see 22, 34] guided by past theoretical and empirical work on college students [also see 20, 23]. We believe that the Belonging Uncertainty measure is the most appropriate to analyze dynamic state-level feelings of belonging especially from a daily diary study (see Walton & Cohen [2007] for theoretical motivations). That said, we also assessed one more state-level belonging measure which asked students, to indicate their agreement with the statement, "Right now, I feel like I belong at [college]" (1 = Strongly disagree, 7 = Strongly agree). Because this measure was phrased differently from all other daily diary measures that elicited retrospective self-reports of belonging and alcohol use (i.e., "yesterday") we do not include this measure in our analysis. That said, when we lagged this measure by 1 to assess patterns of associations between belonging and alcohol use on a day-to-day basis, the results were qualitatively similar and available from the authors on request.

Note that higher values of the *daily belonging* measure denote lower belonging. Hereafter, we refer to this measure as "belonging uncertainty" for clarity and consistency.

**Daily alcohol use.** Each day, participants were asked if they consumed any alcohol; if they reported any use, they were asked follow-up questions regarding the number of drinks consumed. This information was used to create the following daily variables: *Any Alcohol Use* (*1 = Yes, 0 = No*); *Number of Drinks* consumed (measured continuously, including 0). The

results are substantively the same when we analyzed number of drinks only on drinking days (i.e., excluding 0) and available from the authors upon request.

**Sociodemographic factors.** Self-reports were collected to determine self-identified *Gender* (*1 = Woman; 0 = Man*). *Minoritized student status* was measured based on self-reported race/ethnicity. Based on past research on stigmatization and belonging in higher education, Asian, African-American students, Hispanic students, Native American and Pacific Islanders as well as multiracial students were classified as minoritized students in the mixed models (*1 = Minoritized students, 0 = Non-Minoritized)*. The results are substantively the same when we include cross-level interactions separately for Black, Hispanic, Asian, and other race/ethnicity students separately though precision of the effects vary given varying subgroup sample sizes (see S1 Table in the S1 File). *First-generation status* was measured based on student-reported parental education. If neither parent had received a four-year college degree, the student was classified as a first-generation student *(1 = First-generation student; 0 = Continuing-Generation Student)*. Similarly, students who identified as Lesbian, Gay, Bisexual, Transgender, or Queer were classified as an *LGBTQ student* (*1 = LGBTQ Student; 0 = Non-LGBTQ Student*). Given the low sample size of Lesbian, Gay, Bisexual, Transgender, or Queer student subgroups, we do not conduct separate analysis for these subgroups. Even though the SELF study collected information on gender identity and sexual identity in two separate questions, we only have a calculated variable on gender/sexual minority (LGBTQ–yes/no) in our analysis, as approved by this study's IRB.

## Analytic plan

**RQ1: Associations between daily sense of belonging and alcohol use.** To address RQ1, multilevel mixed models were used to examine the within-person association between daily-assessed sense of belonging uncertainty and daily alcohol use behaviors (any alcohol use, total number of drinks, and solitary drinking) across 21 days in our analytical sample. The daily measure of belonging uncertainty was person-mean centered at Level 1 and the centered person-mean measure of belonging uncertainty was included at Level 2 so that person- and day-level effects of belonging uncertainty could be separated. We controlled for the key sociodemographic characteristics described above (gender, minoritized student status, first-generation status, and LGBTQ identity). We included random intercepts and random slopes [40, 41].

**RQ2: Associations between daily sense of belonging and alcohol use by key demographic groups.** To address RQ2, we examined key demographic variables as moderators of the association between daily-assessed sense of belonging and alcohol use outcomes, testing each moderator in separate models. Specifically, we examined cross-level interactions of key sociodemographic factors—gender, minoritized student status, FG status, and LGBTQ status—with daily belonging uncertainty to predict alcohol use outcomes.

## Results

In all, 2,068 participants consented and participated in the Student Engagement, Learning and Flourishing (SELF) survey; of these students, 2,012 participants completed at least 1 day of the 21-day daily diary survey. Students completed an average of 16.8 (SD = 5.7) of the 21 daily surveys, providing a total of 33,722 total person-days. The analytical sample with non-missing values on key measures used in the study is shown in Table 1. While the analytical sample had a slightly larger proportion of female students than the broader university population (68.5% vs. 47.3% in population), the racial-ethnic composition of the sample (7.8% Hispanic, 2.5% non-Hispanic Black, 70.5% non-Hispanic White, 14.4% non-Hispanic Asian, and 4.7% other/mixed races) was largely similar to that of the population of students at this campus (7.7%

Hispanic, 4.3% non-Hispanic Black, 65.2% non-Hispanic White, 6.7% non-Hispanic Asian, and 16.1% other/mixed races). For further details on the Project SELF protocol, please see Lanza and colleagues' protocol [42].

In Table 2, we report descriptive statistics across our key predictor of interest—belonging uncertainty—and two outcomes examining the various facets of drinking behaviors in college —any alcohol use and number of drinks. As shown in Table 2, Non-Hispanic Black students reported the highest mean level of belonging uncertainty (2.114) and Non-Hispanic White the lowest level (1.799). Similarly, we also note that first-generation students and LGBTQ students report higher mean levels of belonging uncertainty (i.e., lower belonging) as compared to the continuing-generation, and non-LGBTQ students, respectively.

**Table 2. Means and standard deviations of key variables (Disaggregated by demographic subgroups).**

| | Between-person level | | | | Within-person level | | | |
|---|---|---|---|---|---|---|---|---|
| | Person-level N | Daily belonging Uncertainty | Any Alcohol Use | Number of drinks | Daily-level N | Daily belonging Uncertainty | Any Alcohol Use | Number of drinks |
| All | 1,913 | 1.831 | 0.148 | 0.656 | 31,422 | 1.804 | 0.151 | 0.670 |
| | | (0.843) | (0.184) | (1.032) | | (1.017) | (0.358) | (1.995) |
| Gender | | | | | | | | |
| Women | 1,309 | 1.861 | 0.148 | 0.585 | 21,910 | 1.834 | 0.152 | 0.598 |
| | | (0.836) | (0.171) | (0.804) | | (1.024) | (0.359) | (1.737) |
| Men | 604 | 1.767 | 0.148 | 0.812 | 9,512 | 1.733 | 0.15 | 0.835 |
| | | (0.857) | (0.209) | (1.392) | | (0.999) | (0.357) | (2.480) |
| Race/Ethnicity | | | | | | | | |
| Non-Hispanic White | 1,349 | 1.799 | 0.171 | 0.778 | 22,736 | 1.776 | 0.172 | 0.780 |
| | | (0.823) | (0.192) | (1.119) | | (1.003) | (0.378) | (2.151) |
| Non-Hispanic Black | 48 | 2.114 | 0.070 | 0.162 | 618 | 2.070 | 0.065 | 0.154 |
| | | (1.035) | (0.125) | (0.323) | | (1.196) | (0.246) | (0.658) |
| Hispanic | 150 | 1.904 | 0.137 | 0.562 | 2,352 | 1.879 | 0.141 | 0.597 |
| | | (0.921) | (0.159) | (0.784) | | (1.113) | (0.348) | (1.860) |
| Non-Hispanic Asian | 276 | 1.861 | 0.071 | 0.279 | 4,278 | 1.846 | 0.071 | 0.279 |
| | | (0.798) | (0.141) | (0.678) | | (0.979) | (0.257) | (1.269) |
| All other race/Multiracial | 90 | 1.947 | 0.106 | 0.414 | 1,438 | 1.873 | 0.111 | 0.432 |
| | | (0.991) | (0.150) | (0.720) | | (1.082) | (0.314) | (1.514) |
| First-generation | | | | | | | | |
| Yes | 324 | 1.900 | 0.151 | 0.567 | 26,212 | 1.891 | 0.149 | 0.562 |
| | | (0.909) | (0.196) | (0.947) | | (1.094) | (0.356) | (1.717) |
| No | 1,589 | 1.817 | 0.148 | 0.675 | 5,210 | 1.786 | 0.152 | 0.692 |
| | | (0.829) | (0.181) | (1.047) | | (1.000) | (0.359) | (2.045) |
| LGBTQ | | | | | | | | |
| Yes | 320 | 2.097 | 0.122 | 0.434 | 5,261 | 2.045 | 0.122 | 0.445 |
| | | (0.886) | (0.152) | (0.667) | | (1.084) | (0.328) | (1.518) |
| No | 1,593 | 1.778 | 0.154 | 0.701 | 26,161 | 1.755 | 0.157 | 0.715 |
| | | (0.825) | (0.189) | (1.085) | | (0.996) | (0.364) | (2.074) |

*Note.* Each cell reports the Mean. SD is in parenthesis. Descriptive statistics of analytic sample for the *any alcohol use* model were reported here. Analytic sample sizes by subgroups vary across models due to missing cases in dependent variables.

Further, we found slightly higher mean rates of drinking among Non-Hispanic White students as compared to students from all other race/ethnicity, and lower drinking among LGBTQ students in comparison to non-LGBTQ students. Results are similar when day of week is controlled for in these models, and available in the S1 File (see S2 Table in the S1 File). The reference day used (i.e., Sunday vs. other days of the week) in the models also do not affect the results (results available upon request).

As shown in Table 3, we examined associations between students' day-level belonging uncertainty and drinking behaviors including cross-level interactions. Within each outcome of interest, the first column shows the baseline model specification (no cross-level interactions included here) to examine RQ1 in the full sample. Across the two outcomes examined, we find consistent patterns. On days when students felt more uncertain about their belonging (than their average), they were less likely to drink and consumed fewer drinks.

Next, to explore moderations by subgroups, columns 2–5 in each panel examines, in separate models, the moderation effects of gender, minoritized student status, FG status, and LGBTQ status on the association between students' belonging uncertainty and each alcohol use outcome. We find that minoritized students were *more* likely to drink and drank more alcohol on days they reported higher belonging uncertainty (i.e., lower belonging). Specifically, we find significant positive cross-level interaction terms of 0.019 and 0.087 for minoritized students and negative main effects of day-level belonging uncertainty of -0.015 and -0.079 overall (see column 3). In other words, on days when minoritized students felt more uncertain about their belonging they drank more often and more drinks overall (as compared to non-minoritized students). We did not observe any other statistically significant cross-level interactions.

## Discussion

Belonging is a key psychosocial determinant of mental health and health behaviors among college students, particularly for women, underrepresented racial/ethnic minorities, first-generation students, and students identifying as LGBTQ during the COVID-19 pandemic [9, 11, 18]. To date, only limited research has examined the construct of belonging as a dynamic process especially as it relates to mental health [25] and alcohol use [29]. To this end, the current study used 21-day diary data from a large, college student sample during the COVID-19 pandemic to assess day-level associations between feelings of belonging and alcohol use outcomes. We also explored whether associations differed by key demographic characteristics given our large analytical sample.

First, we found that on days when students felt more uncertain about their belonging in college than usual (i.e., controlling for person-level belonging uncertainty, which is also negatively related to drinking behaviors), they were less likely to drink and consumed fewer drinks. This negative association might be a result of students seeking out fewer social interactions on days that they feel like they belong less, given that drinking is a very social activity on college campuses. These findings are in line with prior patterns observed in the literature from cross-sectional snapshots that higher levels of belonging might indeed be a risk factor for engaging in drinking behaviors [33].

Next, and most concerningly, we observe that on days that minoritized students feel more uncertain about their belonging, they are more likely to drink. Indeed, we see that the sign and strength of the associations for students from these health disparity populations is significantly different from the full sample results—pointing to the need to examine heterogeneity across important interpersonal dimensions [43]. The sensitivity of the relationship between students' uncertainty about their belonging with their institutions and the likelihood of drinking is a particularly concerning moderation, likely exacerbated during the pandemic. Although we

**Table 3. Mixed models examining associations between belonging uncertainty and alcohol use behaviors.**

| Dependent Variables | Any Alcohol Use | | | | | Total Number of Drinks | | | | |
|---|---|---|---|---|---|---|---|---|---|---|
| **Fixed effects** | (1) | (2) | (3) | (4) | (5) | (1) | (2) | (3) | (4) | (5) |
| Level 1 | | | | | | | | | | |
| Daily Person-centered Belonging Uncertainty | -0.029*** | -0.020*** | -0.015** | -0.027*** | -0.030*** | -0.140*** | -0.121*** | -0.079** | -0.131*** | -0.148*** |
| | (0.003) | (0.007) | (0.006) | (0.004) | (0.004) | (0.019) | (0.037) | (0.036) | (0.021) | (0.022) |
| Level 2 | | | | | | | | | | |
| Average person-mean Belonging Uncertainty | -0.019*** | -0.019*** | -0.019*** | -0.019*** | -0.019*** | -0.122*** | -0.122*** | -0.122*** | -0.122*** | -0.122*** |
| | (0.005) | (0.005) | (0.005) | (0.005) | (0.005) | (0.028) | (0.028) | (0.028) | (0.028) | (0.028) |
| Woman | -0.001 | -0.001 | -0.001 | -0.001 | -0.001 | -0.230*** | -0.230*** | -0.230*** | -0.230*** | -0.230*** |
| | (0.009) | (0.009) | (0.009) | (0.009) | (0.009) | (0.050) | (0.050) | (0.050) | (0.050) | (0.050) |
| Minoritized Student | -0.075*** | -0.075*** | -0.075*** | -0.075*** | -0.075*** | -0.411*** | -0.411*** | -0.411*** | -0.411*** | -0.411*** |
| | (0.009) | (0.009) | (0.009) | (0.009) | (0.009) | (0.051) | (0.051) | (0.051) | (0.051) | (0.051) |
| First generation Student | 0.010 | 0.010 | 0.010 | 0.01 | 0.01 | -0.048 | -0.048 | -0.048 | -0.048 | -0.048 |
| | (0.011) | (0.011) | (0.011) | (0.011) | (0.011) | (0.062) | (0.062) | (0.062) | (0.062) | (0.062) |
| LGBTQ | -0.028** | -0.028** | -0.028** | -0.028** | -0.028** | -0.199*** | -0.199*** | -0.199*** | -0.199*** | -0.199*** |
| | (0.011) | (0.011) | (0.011) | (0.011) | (0.011) | (0.062) | (0.062) | (0.062) | (0.062) | (0.062) |
| Cross-level interactions | | | | | | | | | | |
| Daily Belonging Uncertainty x Woman | | -0.012 | | | | | -0.027 | | | |
| | | (0.008) | | | | | (0.043) | | | |
| Daily Belonging Uncertainty x Minoritized | | | 0.019*** | | | | | 0.087** | | |
| | | | (0.008) | | | | | (0.042) | | |
| Daily Belonging Uncertainty x First generation | | | | -0.013 | | | | | -0.057 | |
| | | | | (0.009) | | | | | (0.051) | |
| Daily Belonging Uncertainty x LGBTQ | | | | | 0.005 | | | | | 0.037 |
| | | | | | (0.009) | | | | | (0.048) |
| Random effects | | | | | | | | | | |
| Slope variance | 0.002*** | 0.002*** | 0.002*** | 0.002*** | 0.002*** | 0.079*** | 0.079*** | 0.079*** | 0.079*** | 0.079*** |
| | (0.001) | (0.001) | (0.001) | (0.001) | (0.001) | (0.017) | (0.017) | (0.017) | (0.017) | (0.017) |
| Intercept variance | 0.023*** | 0.023*** | 0.023*** | 0.023*** | 0.023*** | 0.767*** | 0.767*** | 0.767*** | 0.767*** | 0.767*** |
| | (0.001) | (0.001) | (0.001) | (0.001) | (0.001) | (0.032) | (0.032) | (0.032) | (0.032) | (0.032) |
| Residual variance | 0.102*** | 0.102*** | 0.102*** | 0.102*** | 0.102*** | 3.109*** | 3.109*** | 3.109*** | 3.109*** | 3.109*** |
| | (0.001) | (0.001) | (0.001) | (0.001) | (0.001) | (0.026) | (0.026) | (0.026) | (0.026) | (0.026) |
| N | 31422 | 31422 | 31422 | 31422 | 31422 | 31402 | 31402 | 31402 | 31402 | 31402 |

*Note.* Standard errors in parentheses.

* $p < .10$

** $p < .05$

*** $p < .01$.

interpret this effect with caution given sample size differences across various minoritized sub-groups, we believe these trends warrant further attention that future research from more diverse, large, samples should focus on.

Finally, minoritized, first-generation, and LGBTQ students show slightly higher belonging uncertainty even on a day-to-day basis, largely replicating prior work [20, 24]. We also observe larger within-person variations in belonging uncertainty for these groups. These results are consistent with prior theory and empirical research, that minoritized students, especially in predominantly white institutions are often exposed to daily slights and adversities that make

them question their day-to-day belonging [22, 34]. Our study documents that larger variations are also found among other health-disparity populations—such as LGBTQ students—that warrants further research.

Campus closure, stricter norms of gathering during the pandemic, etc. could have contributed to the overall low levels of social drinking that we might be capturing here. That said, students reporting high belonging uncertainty (i.e., lower belonging) may have had fewer opportunities to socialize with others, when drinking is more common and therefore drink less. This interpretation highlighted by Neighbors et al. [29] in their study aligns somewhat with our study findings. In other words, lower likelihood of drinking and drinking less on days when students feel more uncertain about their belonging, are positive outcomes overall given the college population we are studying. However, if these are proxies for social isolation, especially for students from historically, marginalized sociodemographic subgroups in college, future research and outreach/institutional programming efforts must pay close attention to these patterns.

## Limitations

We acknowledge four key limitations of our study. First, our college student sample is from a single, large, 4-year, residential, public university that likely limits the generalizability of our findings. We therefore encourage other universities to conduct similar studies using similar measures of belonging and alcohol use to better understand key contextual moderators. Second, even though we use a rigorous day- and person-level longitudinal, mixed model analysis, we note that the associations we highlight are not causal. We cannot easily address reverse causation—i.e., whether students drink more or more often to feel like they belong on campus or vice versa—with our research design/models. Third, despite our large sample size (both person- and day-level) overall in the analytic sample, disaggregation by finer-grained race/ethnic, or LGBTQ subgroups is constrained by statistical power. For example, the LGBTQ categorization might mask potential differences by gender- and/or sexual-identity categories that we have not been able to tease out adequately. Finally, although we use an extensively studied measure of belonging uncertainty in college student samples [see 20, 22, 23, 34], we cannot rule out the limitations related to the measurement challenges inherent in belonging assessments used in the literature [44].

## Conclusion

Our findings point towards the continued need for more data and research from a variety of institutions as students continue to navigate their college experiences amidst the pandemic. Several studies have found that students from minoritized backgrounds often report lower levels of belonging on days when they experience negative, interpersonal interactions. Interventions that provide more adaptive interpretations of struggles during the transition to college, have buffered students from making global assessments about their sense of belonging, which results in a self-reinforcing cycle of positive outcomes for those students [20, 22]. Such interventions can be further adapted to reduce the likelihood of increased drinking as a strategy for gaining acceptance or feeling more connected.

Despite the limitations identified in the prior section, feelings of isolation and uncertainty about college students' belonging and connections with peers and colleges might have an unintended consequence—drinking more and more often—particularly for vulnerable, health-disparity populations, that colleges should continue to monitor. In-depth studies of belonging and wellbeing with innovative data collection efforts such as the SELF that includes daily diary assessments help further elucidate the underlying, dynamic processes. We believe that results

from these findings will thus be invaluable for informing institutional prevention and outreach efforts, answering national calls to research and action to improve students' overall mental health and wellbeing [45].

## Supporting information

**S1 Checklist. Human participants research checklist.**
(DOCX)

**S1 File. Supplemental materials for "College students' sense of belonging and alcohol use amidst COVID-19: Evidence from a 21-day daily diary study".**
(DOCX)

## Acknowledgments

We thank Courtney Whetzel and Sandesh Bhandari for their help with survey administration, project management, and data management efforts. All remaining errors are our own.

## Author Contributions

**Conceptualization:** Maithreyi Gopalan, Ashley Linden-Carmichael, Stephanie Lanza.

**Formal analysis:** Maithreyi Gopalan, Jilli Jung, Chiang Shou-Chun.

**Funding acquisition:** Stephanie Lanza.

**Investigation:** Maithreyi Gopalan, Jilli Jung, Chiang Shou-Chun, Ashley Linden-Carmichael, Stephanie Lanza.

**Supervision:** Ashley Linden-Carmichael, Stephanie Lanza.

**Visualization:** Jilli Jung.

**Writing – original draft:** Maithreyi Gopalan.

**Writing – review & editing:** Maithreyi Gopalan, Ashley Linden-Carmichael, Stephanie Lanza.

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
