## [Decision Letter · Decision Letter 0]

21 May 2024

PONE-D-23-38054College Students’ Sense of Belonging and Alcohol Use amidst COVID: Evidence from a 21-day Daily Diary StudyPLOS ONE

Dear Dr. Gopalan,

Thank you for submitting your manuscript to PLOS ONE. After careful consideration, we feel that it has merit but does not fully meet PLOS ONE’s publication criteria as it currently stands. Therefore, we invite you to submit a revised version of the manuscript that addresses the points raised during the review process.

**ACADEMIC EDITOR: You have done an interesting study. However, some modifications are expected to add clarity to the study for the readers or research community. Please revise the manuscript thoroughly as per the comments raised by the authors..**

We look forward to receiving your revised manuscript.

Kind regards,

Yadeta Alemayehu

Academic Editor

PLOS ONE

2. In the online submission form you indicate that your data is not available for proprietary reasons and have provided a contact point for accessing this data. Please note that your current contact point is a co-author on this manuscript. According to our Data Policy, the contact point must not be an author on the manuscript and must be an institutional contact, ideally not an individual. Please revise your data statement to a non-author institutional point of contact, such as a data access or ethics committee, and send this to us via return email. Please also include contact information for the third party organization, and please include the full citation of where the data can be found.

3. Please ensure that you include a title page within your main document. You should list all authors and all affiliations as per our author instructions and clearly indicate the corresponding author.

Additional Editor Comments:

Please revise the reference of the manuscript as per the standard Referencing style of PLOS ONE journal: https://journals.plos.org/plosone/s/submission-guidelines#loc-references.Incorporate some main methodologies and findings in the abstract section rather than focusing mainly on the conclusion and public Health Significance Statements.Although the conclusion subsection is specified in the abstract section it is not a sub section in the main document so its better to add this subsection independently.

Reviewers' comments:

Reviewer's Responses to Questions

**Comments to the Author**

1. Is the manuscript technically sound, and do the data support the conclusions?

Reviewer #1: Yes

Reviewer #2: Yes

Reviewer #3: Yes

Reviewer #4: Yes

Reviewer #5: Partly

2. Has the statistical analysis been performed appropriately and rigorously? 

Reviewer #1: Yes

Reviewer #2: Yes

Reviewer #3: Yes

Reviewer #4: I Don't Know

Reviewer #5: Yes

3. Have the authors made all data underlying the findings in their manuscript fully available?

Reviewer #1: No

Reviewer #2: Yes

Reviewer #3: No

Reviewer #4: Yes

Reviewer #5: No

4. Is the manuscript presented in an intelligible fashion and written in standard English?

Reviewer #1: Yes

Reviewer #2: Yes

Reviewer #3: Yes

Reviewer #4: Yes

Reviewer #5: Yes

5. Review Comments to the Author

Reviewer #1: Review PONE-D-23-38054

Overall remarks:

This article presents the relationship between daily sense of belonging to a campus for college students and their alcohol consumption during the COVID-19 pandemic. The authors demonstrated that more uncertainty regarding sense of belonging to the college campus is related to lower alcohol consumption. Underrepresented racial minorities and student identifying to the LGBTQ community reported more uncertainty with their belonging, which was related to higher alcohol use.

This article is overall well written and uses appropriate statistical methods to respond to their research questions. However, it is not publishable in its current state. Throughout this article, slight modifications in the phrasing and corrections could improve dramatically its clarity. Also, content belonging in the results section is presented in the methods and should be moved. Furthermore, methodological issues are not addressed properly and could benefit from transparency and a deeper discussion in the methods or in the limitations section.

Finaly this article seem relevant and could be of interest for the readers of PLOS One.

Abstract:

The abstract is short and present a good overview of the content of this manuscript. However, slight modifications could greatly improve its quality.

- What to you refer to when talking about sense of belonging? To what institution or group’s belonging are you measuring in this article?

- The phrasing of the result section could be improved as the first sentence can be hard to understand at first glance.

Introduction:

The introduction offers a good overview of the current literature and introduces well the concepts used in this manuscript. However, it could benefit from a little introduction to the definition of the sense of belonging and of measures often used to measure it. This could offer a good ground for comparison with the measure used in this project and

- This sentence in the second paragraph is not clear: “Specifically, while some studies found that college students reported heavier use during relative to prior to the pandemic […]” are you referring to before the pandemic? At the beginning of the pandemic?

- The justification for the inclusion of sexual diversity, gender diversity and racial diversity is short, could you add a bit more context for the audience to better understand why these dimensions are important? Most of these populations are at risk for mental health disorder, but is it the case for alcohol use and abuse?

- This affirmation at page 4 needs a citation: “On the prevention side, a key emerging protective factor against adverse mental health outcomes—such as anxiety and depression, which is often a precursor to alcohol use […]”

- This sentence in page 4 is a bit of an overstatement: “Students’ sense of belonging with their campus, often considered a fundamental human motivation (Baumeister & Leary, 1995) […]” the provided reference addresses sense of belonging at large, not something as specific as belonging to a specific campus. You can extend the importance of belonging to a college campus but please rephrase that. The rest of the sentence is really good and could support this extension.

- This sentence in page 4 has no citation to support it: “For example, college students who reported feeling a strong sense of belonging with their institution prior to the onset of the pandemic reported lower levels of depressive symptoms and anxiety during the pandemic, even after controlling for their baseline (pre-COVID) depressive and anxiety symptoms.”

- This sentence in page 5 is very interesting: “First, there is consistent evidence of sociodemographic differences by race/ethnicity, gender, and first-generation college student status in students’ sense of belonging.” but could you give a little bit more information about those results as this could offer already an idea of the direction of your results despite its longer distance between measures.

- This section could benefit greatly from a slight introduction to the concept of belonging, a definition and a quick presentation of the measures often used to quantify this concept. The addition of a footnote in the methods section offers an important clarification of the method used but personally raised more questions regarding this measure, it’s quality and the actual concept measured here.

Methods:

The outline of this section is good, and it is overall well written. Nonetheless the presentation of measures should be dramatically improved for transparency and address eventual methods limitations in the study design and/or statistical analyses.

- I appreciate the transparency regarding ethical requirements in the PLOS ONE form, but could you please add a small statement in the methods section stating the university that you refer to in this sentence page 7: “The university’s Institutional Review Board approved the current study.”

- Descriptive statistics, including table 1 and the paragraph following it before the Measures section should belong in the results section (i.e. “While the analytical sample had a slightly larger proportion of female students than the broader university population (68.5% vs. 47.3% in population), the racial-ethnic composition of the sample (7.8% Hispanic, 2.5% non-Hispanic Black, 70.5% non-Hispanic White, 14.4% non-Hispanic Asian, and 4.7% other/mixed races) was largely similar to that of the population of students at this campus (7.7% Hispanic, 4.3% non-Hispanic Black, 65.2% non-Hispanic White, 6.7% non-Hispanic Asian, and 16.1% other/mixed races). For further details on the Project SELF protocol, please see Lanza and colleagues’ protocol (under review).”).

- The measure of belonging uncertainty is interesting as it seems to capture a large portion of the internal perspective of the student on its situation. It is nonetheless very limited as belonging to a college can have many layers such as belonging to specific groups within this institution, agreeing with this college’s administrative choices, etc. I do not reject this measure but as evoked in the introduction section, I would really like to see more regarding the definition of sense of belonging and addressing this small variable’s eventual limitations in this regard.

- The measure of alcohol used alone or in a social setting is not presented here but included in the analytic plan which is rather confusing.

- The sociodemographic factors used are on one hand relevant and interesting but on the other very limited and incoherent, especially regarding the LGBTQ status. You add a measure regarding LGBTQ status but only offer two choices regarding gender, which is rather antithetic. The LGBTQ status refer to sexual and gender-diversity which are very different dimensions that can have major differences in the observed dynamics. When looking at the provided footnote, this classification as LGBTQ or non-LGBTQ seems to be rather a statistical choice rather than a methodological flaw. This is the same thing for racial groups. The measures in the methods section should be presented raw, as the variables were collected in the design, not as you chose to include it in your statistical analysis. The statistical choices should be provided after the presentation of the measures, so the overall design is totally transparent and your statistical choices too. The lack of diversity in your final sample can be interpreted as a limitation and your choice to group them to curb this limitation can be relevant but transparency is extremely important here. Therefore, I highly recommend reviewing thoroughly this section for us to understand if the limitation regarding sex and gender diversity is a methodological flaw in the study design, or rather a voluntary choice to balance statistical power and your research objectives.

- I appreciate the inclusion of an appendix table to present dynamics in racial subgroups.

- The analytic plan is clear and seem appropriate to respond to your questions.

Results:

The results are overall well presented and seem complete. However, as your sense of belonging variable is reversed, it is a bit more complicated to understand the direction of results.

Discussion and conclusion:

- Did you control for the day of the week when measures were taken? As social events typically occur during weekends, this would be extremely important to measure as students going to an event organized by the college’s student associations for example could feel a higher belonging and consume more alcohol, which could mostly drive your observed effect.

- This sentence needs a citation: “Finally, women, URM, first-generation, and LGBTQ students show slightly higher belonging uncertainty even on a day-to-day basis, largely replicating prior work.”

- The limitation section should underline the limitations regarding LGBTQ and URM groups being the lack of fine measures of LGBTQ status and the grouping of URM groups due to small size as these can have an important influence on results and should be addressed in further studies.

Reviewer #2: Thank you very much for giving me the opportunity to review this article. I found the topic quite interesting, and a very important issue to consider when working with college students. This kind of investigations are necessary to improve the passage of undergraduates through the college years. Although it is very well documented, I have some concerns with the way in which the study has been approached. I have the impression that the approach and the analyses do not coincide with the discussion of the results, as if there were two different objectives.

Introduction:

p. 3 “Given that major differences exist in the impact of COVID-19 on health outcomes by gender, underrepresented racial and ethnic minorities status (URM), first-generation status, and LGBTQ status” How is it? It would be beneficial for the introduction to specify the patterns and the results of the different studies, to better understand which the differences are.

It would be interesting to describe how the situation was in 2021 when the study was implemented. There are countries, such as Spain, where the measures and policies were very strict by then (curfews, limitations on being at the street, limitations on group reunions, etc.). These specific situations may have had an impact on the alcohol use tendencies in college students. If students had to share more time in campus because of some specific policy, maybe it led to a higher sense of belonging, and vice versa. For example, during 2021, in some Spanish universities, students only had to be one of every three weeks present in classes, with the 2 other weeks following remote lessons. This situation irremediably must affect the sense of belonging, particularly in those that 2021 was their first college year. The authors include some reflections in the discussion about this issue, but I feel that it would be positive to address it in the introduction also.

Method and analyses

It is no clear for me if the research questions included where or with whom the participants drank, because it is not the same to drink alone that to meet some friends or colleagues to have some drinks. I believe that this is also an interesting point to consider in this investigation, because I assume that the way and the people you drink with affect in any way to the sense of belonging.

Discussion

Also, although the discussion is very well conducted, and the analyses show interesting results, I got confused during the reading of the manuscript because I do not know if the results were as expected or were completely different. I know that the research questions were mostly descriptive, and that the study design responds to them, but for me was unexpected the result “a higher uncertainty in belonging is related to a less likely of consumption of alcohol”. It is true that the authors explain this result in the discussion, but in the introduction, I could not find information related to this possible result, or to the contrary. In fact, it seems that the main result (that URM population had the opposite behavior than the other groups) has nothing to do with the objective of the study (because it is no mentioned that this result can be achieved).

It is clear that alcohol consumption can be problematic in university students, and that university institutions must address this issue; however, I believe that the future lines of investigation are also scarce. There is potential in the study results that I believe is not being addressed. On the one hand, to implement measures of welcome, of belonging that do not involve alcohol consumption (since, according to the study, the greater the feeling of belonging, the greater the alcohol consumption). On the other hand, what can be done with minority populations (who happen to be the ones who consume more alcohol when they feel less belonging)?

For all these reasons, I consider that the article should be rewritten, emphasizing in the introduction the different points that I have tried to highlight in the review.

Reviewer #3: Overall, the manuscript provides valuable insights into the linkages between a key psychosocial factor—students’ sense of belonging in college—and their alcohol use patterns amidst COVID. The manuscript's findings hold significant value for mental health care and public health services. It is well-written and organized, with a clear introduction that sets the context for the study. However, implementing the recommendations would further improve the clarity, robustness, and applicability of the findings.

General recommendation:

1. Although the language is generally clear, improving sentence structure, selecting better words, and ensuring coherence could enhance readability. Additionally, it is recommended to proofread for grammatical and typographical errors.

2. The reference style does not meet the journal's requirements, and it should adhere to the Vancouver style.

More specific recommendation:

introduction:

The introduction is well-written, concise, and clear, effectively highlighting the gap in the literature and explaining the aim of the present study. The heading for the current study at the end of the introduction is provided very effectively.

Method:

In this section, it is advisable to introduce the study's methodology before delving into the sampling and population. This could include specifying whether the study is qualitative or quantitative, descriptive or applied, and whether it involves multivariate correlation analysis, among other relevant details.

Participant’s part is clear, however:

Table 1, labeled as "Descriptive Statistics" in this section, should be renamed to avoid misleading the reader. This is because Table 2, titled as "Descriptive Statistics of Participants," also appears in the results section, and it's important to distinguish between the two.

The paragraph that begins with "In all, 2,068 participants consented and participated in the Student Engagement…" should be titled as "Procedure" as a subheading in the Method section.

Results:

The results section offers a comprehensive presentation of relevant statistical analyses. The tables effectively convey important data necessary for interpreting the results, and the author skillfully avoids redundancy and unnecessary elaboration within the tables.

However, the titles of the tables need to be more specific and clear for readers.

Discussion:

The discussion part effectively explores the study findings and comparing them with previous research. It appropriately addresses the limitations of the study and suggests avenues for future research.

It is recommended that the last paragraph of the discussion part be titled "Conclusion" to provide clarity for readers.

In the last sentence of paragraph 2, which starts with "These findings are in line with prior patterns observed in the literature from cross-sectional…", the reference is missing.

Reviewer #4: Greetings and Regards,

I am so grateful to had an opportunity to read this valuable manuscript.

After a detailed review of the submitted manuscript, below are some notes that may help improve readers' understanding.

1. It is suggested to mention the sampling method.

2. What exclusion criteria were considered?

3. It is suggested to briefly mention ethical considerations and how to obtain consent.

4. Were psychometric criteria such as face validity considered before choosing the question to measure the sense of belonging? How are you sure that the said question has the ability to measure this concept in the population?

5. The second paragraph of the discussion needs some refrences.

6. In the title of the article, it is mentioned that the study was conducted during the Covid pandemic, but there is not much mention of the role of Covid in the level of fear, anxiety and isolation of people, and especially in the discussion part, there is not much mention of it and similar studies.

Reviewer #5: An operational definition of 'belonging' could be included.

The authors may qualify "public university in the Northeast" of so-and-so country under participant inclusion criteria.

It was not very clear whether the authors intended to and/or found drinking alcohol to be indicative of belonging, or if drinking was a coping mechanism for belonging uncertainty.

6. PLOS authors have the option to publish the peer review history of their article (what does this mean?). If published, this will include your full peer review and any attached files.

Reviewer #1: No

Reviewer #2: No

Reviewer #3: No

Reviewer #4: No

Reviewer #5: No

---

## [Author Response · Author response to Decision Letter 0]

8 Jul 2024

Please find response letter attached in the files (Filename: ResponseLetter_R1_Submitted.docx) that includes our detailed responses to reviewer concerns.

---

## [Decision Letter · Decision Letter 1]

6 Aug 2024

PONE-D-23-38054R1College Students’ Sense of Belonging and Alcohol Use amidst COVID: Evidence from a 21-day Daily Diary StudyPLOS ONE

Dear Dr. Gopalan,

Thank you for submitting your manuscript to PLOS ONE. After careful consideration, we feel that it has merit but does not fully meet PLOS ONE’s publication criteria as it currently stands. Therefore, we invite you to submit a revised version of the manuscript that addresses the points raised during the review process.

**ACADEMIC EDITOR: **I want to appreciate the modification you made on the first draft of your article however some minor revisions are still required. Furthermore, please consider the comments provided by reviewers that focuses on the clarification of methodologies and the result. Please submit your revised manuscript by Sep 20 2024 11:59PM. If you will need more time than this to complete your revisions, please reply to this message or contact the journal office at plosone@plos.org. Please include the following items when submitting your revised manuscript:A rebuttal letter that responds to each point raised by the academic editor and reviewer(s). You should upload this letter as a separate file labeled 'Response to Reviewers'.A marked-up copy of your manuscript that highlights changes made to the original version. You should upload this as a separate file labeled 'Revised Manuscript with Track Changes'.An unmarked version of your revised paper without tracked changes. You should upload this as a separate file labeled 'Manuscript'.If applicable, we recommend that you deposit your laboratory protocols in protocols.io to enhance the reproducibility of your results. Protocols.io assigns your protocol its own identifier (DOI) so that it can be cited independently in the future. For instructions see: https://journals.plos.org/plosone/s/submission-guidelines#loc-laboratory-protocols. Additionally, PLOS ONE offers an option for publishing peer-reviewed Lab Protocol articles, which describe protocols hosted on protocols.io. Read more information on sharing protocols at https://plos.org/protocols?utm_medium=editorial-email&utm_source=authorletters&utm_campaign=protocols.

We look forward to receiving your revised manuscript.

Kind regards,

Yadeta Alemayehu

Academic Editor

PLOS ONE

Journal Requirements:

Reviewers' comments:

Reviewer's Responses to Questions

**Comments to the Author**

1. If the authors have adequately addressed your comments raised in a previous round of review and you feel that this manuscript is now acceptable for publication, you may indicate that here to bypass the “Comments to the Author” section, enter your conflict of interest statement in the “Confidential to Editor” section, and submit your "Accept" recommendation.

Reviewer #1: (No Response)

Reviewer #2: All comments have been addressed

Reviewer #3: All comments have been addressed

Reviewer #5: (No Response)

2. Is the manuscript technically sound, and do the data support the conclusions?

Reviewer #1: Yes

Reviewer #2: Yes

Reviewer #3: Yes

Reviewer #5: Yes

3. Has the statistical analysis been performed appropriately and rigorously? 

Reviewer #1: Yes

Reviewer #2: N/A

Reviewer #3: Yes

Reviewer #5: Yes

4. Have the authors made all data underlying the findings in their manuscript fully available?

Reviewer #1: No

Reviewer #2: Yes

Reviewer #3: No

Reviewer #5: Yes

5. Is the manuscript presented in an intelligible fashion and written in standard English?

Reviewer #1: Yes

Reviewer #2: Yes

Reviewer #3: Yes

Reviewer #5: Yes

6. Review Comments to the Author

Reviewer #1: Review #2 PONE-D-23-38054

Overall, this revised version of this manuscript addressed correctly most of the comments provided by reviewers. I want to thank the authors for their work, their thorough responses to comments, and I believe they improved greatly the quality of this manuscript. However, some modifications are still needed as some aspects have not totally been addressed and some small modifications are needed to ease reading flow and overall comprehension.

I did not identify this earlier, but I believe that the right way to state the COVID pandemic is to specify the COVID-19 pandemic as coronaviruses were present before 2020 and induced epidemics before. Therefore, to properly identify this work in its historical context, it is needed to add the -19 as it refers to the year 2019.

Abstract:

This sentence in the result section is long and unclear: “Results revealed that on days when students felt more uncertain about their belonging (i.e., low belonging) with their colleges, they were less likely to drink, and drink less overall even if they do when back on campus after pandemic-related college closures ended.” Please change the phrasing to clarify it’s meaning. Here is a suggestion: Results revealed that on days when students felt more uncertainty about their sense of belonging to their college (i.e., low belonging), they were less likely to drink, and drink less overall. This effect was also present when students went back on campus after pandemic-related college closures ended indicating a limited effect of measures taken to curb the spread of COVID-19 on the relation between sense of belonging and alcohol consumption.

Introduction:

Overall, this section has been dramatically improved. I appreciate the inclusion of clarifications regarding sense of belonging, as well as inclusion of more detailed presentation of the relevance of including sexual diversity, gender diversity, and racial diversity in your paper. Further comments are only superficial aspects to ease reading flow and clarification for potential lay audiences.

Please add some context in the introduction, especially when using references that are specific to the USA. For example, 4-year universities are a US-specific form of post-secondary education structure that are not necessarily present in other countries. Furthermore, what do you mean by “[…] highest rates of drinking”? Are these students more prone to drinking in general? I suggest rephrasing this to reflect the USA-specific status of the study cited for this sentence and clarify what is meant by highest rates of drinking.

Please identify typos and add commas when necessary. For example, this sentence needs a comma after pandemic: “Lastly, during the pandemic individuals who identify as LGBTQ+ have been at disproportionate risk for mental health concerns and substance use (15,16).”

This sentence looks like the long dash is used as a parenthesis: “On the prevention side, a key emerging protective factor against adverse mental health outcomes—such as anxiety and depression, which often co-occur with alcohol use (e.g., 17), was students’ sense of belonging with their university (18).” If so, please close it.

Some quote marks are not closed: (e.g., “I feel like I belong at [college]; “How […]).

Methods:

Here, also, most comments have been properly addressed or responded. However, two aspects still need to be addressed as they have been partially corrected.

I still believe that descriptive statistics of the sample (as presented in table 1) is a result per se and therefore belongs in the results section. Methods section should only contain information on the questionnaires/measures taken and superficial information on the participants such as recruitment procedure, criteria of inclusion and exclusion to address the “How do you respond to your research question?”. Furthermore, the second paragraph of the “procedure” section, especially starting at: “While the analytical sample […]” also is a presentation of results and do not reflect the recruitment procedure nor methods. I’ll let the authors decide if they want to change this or not.

You addressed partially my comment regarding the LGBTQ question but did not clarify how this status was measured in the first place. Is it a simple binary question “Do you identify as LGBTQ? Yes/No” or was it measured differently? I appreciate the addition of a statement in the discussion section, but this is still not clear in the methods as you still do not clearly state how the measures are taken, nor it is detailed in the table 1. Again, as stated in my previous review, I could understand and agree with such a grouped classification, but your LGBTQ sample is not small with almost 17% of the sample and sexual/gender diversity are different concepts that could present very different realities. Therefore, more justification is needed here if the initial measure was not a binary question. As race/ethnicity is presented in a more stratified fashion, I want to be able to see which aspect of the LGBTQ acronym is so small that you need to group them together. Furthermore, maybe a stratification splitting sexual diversity vs gender diversity could be relevant to reflect and address the heterogeneity in the LGBTQ acronym.

Results:

No problem here.

Discussion/conclusion:

In this section, comments have been mostly addressed too but slight modifications are required.

I believe the limitations section should be before the conclusion as the conclusion is supposed to include all aspects of the article, whether positive or negative. Furthermore, the conclusion should be updated to reflect identified limitations and englobe everything. No need to expand on limitations but a quick sentence acknowledging them and counterbalance it with the strong aspects of your work would be sufficient.

I appreciate the inclusion of identified limitations. However, I will reiterate the fact that aggregation by LGBTQ status is not justified enough here, and the limitation do not address in detail the differences inherent to the large umbrella term of LGBTQ. A simple statement acknowledging this might be sufficient, but it is still needed here.

I appreciate that you addressed my comment about controlling for the day of the week and added a table to reflect this but maybe adding a little sentence in the results/discussion section could help readers identify faster that you addressed this question already. I saw the note under the Table 3, but it is rather discrete. Furthermore, why using Sunday as the reference? Just a small rationale justifying this choice would be enough I believe.

I realized that appendix table A1 has no title. Please add one.

Reviewer #2: I thank the authors for their efforts to improve their article and to answer and discuss all the issues I raised in the first review.

Reviewer #3: Appreciate to the author for incorporating all the revisions based on the reviewer's comments and providing explanations for each change. The revised manuscript demonstrates improved design and writing, with a clear and cohesive introduction and discussion section. Overall, the revised manuscript meets the criteria for acceptance.

Reviewer #5: The authors have not adequately addressed the comments raised in the previous round of review. I request the authors to clearly state things like definitions, sources, references, citations, and so forth that other reviewers and I have suggested. Otherwise, it is difficult to okay the submission for publication.

7. PLOS authors have the option to publish the peer review history of their article (what does this mean?). If published, this will include your full peer review and any attached files.

Reviewer #1: No

Reviewer #2: No

Reviewer #3: No

Reviewer #5: No

---

## [Author Response · Author response to Decision Letter 1]

28 Aug 2024

Document (ResponseLetter_R2.docx) that includes detailed response to all reviewer comments attached.

---

## [Editor Report · Decision Letter 2]

2 Sep 2024

College Students’ Sense of Belonging and Alcohol Use amidst COVID-19: Evidence from a 21-day Daily Diary Study

PONE-D-23-38054R2

Dear Dr. Maithreyi,

We’re pleased to inform you that your manuscript has been judged scientifically suitable for publication and will be formally accepted for publication once it meets all outstanding technical requirements.

Kind regards,

Yadeta Alemayehu

Academic Editor

PLOS ONE

---

## [Editor Report · Acceptance letter]

19 Nov 2024

PONE-D-23-38054R2 

PLOS ONE

Dear Dr. Gopalan, 

I'm pleased to inform you that your manuscript has been deemed suitable for publication in PLOS ONE. Congratulations! Your manuscript is now being handed over to our production team.

Kind regards, 

on behalf of

Mr. Yadeta Alemayehu 

Academic Editor

PLOS ONE